# SMYD3 Modulates AMPK-mTOR Signaling Balance in Cancer Cell Response to DNA Damage

**DOI:** 10.3390/cells12222644

**Published:** 2023-11-17

**Authors:** Martina Lepore Signorile, Paola Sanese, Elisabetta Di Nicola, Candida Fasano, Giovanna Forte, Katia De Marco, Vittoria Disciglio, Marialaura Latrofa, Antonino Pantaleo, Greta Varchi, Alberto Del Rio, Valentina Grossi, Cristiano Simone

**Affiliations:** 1Medical Genetics, National Institute of Gastroenterology-IRCCS “Saverio de Bellis” Research Hospital, 70013 Castellana Grotte, Italy; martina.lepore@irccsdebellis.it (M.L.S.); elisabetta.dinicola@irccsdebellis.it (E.D.N.); candida.fasano@irccsdebellis.it (C.F.); giovanna.forte@irccsdebellis.it (G.F.); katia.demarco@irccsdebellis.it (K.D.M.); vittoria.disciglio@irccsdebellis.it (V.D.); m.latrofa1995@outlook.it (M.L.); antonino.pantaleo@irccsdebellis.it (A.P.); valentina.grossi@irccsdebellis.it (V.G.); 2Institute for Organic Synthesis and Photoreactivity (ISOF), National Research Council of Italy (CNR), 40129 Bologna, Italy; greta.varchi@isof.cnr.it (G.V.); alberto.delrio@isof.cnr.it (A.D.R.); 3Innovamol Consulting Srl, 41126 Modena, Italy; 4Medical Genetics, Department of Precision and Regenerative Medicine and Jonic Area (DiMePRe-J), University of Bari Aldo Moro, 70124 Bari, Italy

**Keywords:** SMYD3, AMPK, mTOR, DNA damage, gastrointestinal cancer, breast cancer

## Abstract

Cells respond to DNA damage by activating a complex array of signaling networks, which include the AMPK and mTOR pathways. After DNA double-strand breakage, ATM, a core component of the DNA repair system, activates the AMPK-TSC2 pathway, leading to the inhibition of the mTOR cascade. Recently, we showed that both AMPK and mTOR interact with SMYD3, a methyltransferase involved in DNA damage response. In this study, through extensive molecular characterization of gastrointestinal and breast cancer cells, we found that SMYD3 is part of a multiprotein complex that is involved in DNA damage response and also comprises AMPK and mTOR. In particular, upon exposure to the double-strand break-inducing agent neocarzinostatin, SMYD3 pharmacological inhibition suppressed AMPK cascade activation and thereby promoted the mTOR pathway, which reveals the central role played by SMYD3 in the modulation of AMPK-mTOR signaling balance during cancer cell response to DNA double-strand breaks. Moreover, we found that SMYD3 can methylate AMPK at the evolutionarily conserved residues Lys411 and Lys424. Overall, our data revealed that SMYD3 can act as a bridge between the AMPK and mTOR pathways upon neocarzinostatin-induced DNA damage in gastrointestinal and breast cancer cells.

## 1. Introduction

Cells are exposed daily to endogenous and exogenous sources of DNA damage [1]. Endogenous factors inducing DNA lesions include the production of reactive oxygen, nitrogen, and carbonyl species, byproducts of lipid peroxidation, alkylating agents, estrogen, and cholesterol metabolites, as well as spontaneous hydrolysis of DNA under physiological conditions, leaving non-coding apurinic/apyrimidinic abasic sites [2,3]. On the other hand, exogenous factors comprise ultraviolet (UV) light, ionizing radiation (IR), genotoxic chemicals, and carcinogens that are inhaled or ingested [3]. DNA damage can be classified into different types, including single-base alterations (depuration, deamination, base alkylation), two-base alterations (UV-induced pyrimidine dimers), single-strand breaks, and double-strand breaks (DSBs), with DSBs being the most significant and lethal damage since they do not leave an intact complementary strand to be used as a template for DNA repair [4]. DSBs are frequently caused by environmental exposure to irradiation or chemical agents such as the radiomimetic drug neocarzinostatin (NCS) [5]. In order to deal with DNA damage, cells can activate several cellular processes, including progression through cell cycle checkpoints, DNA repair, transcriptional adjustment, and eventually apoptosis [6]. Recently, several findings revealed the involvement of the mTOR complex 1 (mTORC1) pathway in DNA damage response (DDR). mTORC1 is a multiprotein complex that comprises mTOR, RAPTOR, and mLST8. mTOR, like ATM, belongs to the phosphatidylinositol 3-kinase-related kinase (PIKK) family, a group of proteins playing a major role in DNA damage signaling [6]. As a result of DNA damage, active ATM phosphorylates several proteins, including CHK2 and p53, and coordinates cell cycle arrest and DNA repair [7]. Prolonged treatment with DSB-inducing drugs results in ATM-dependent mTOR inhibition [8]. Importantly, ATM transduces its signal to mTOR through the AMPK-TSC2 pathway [9]. Indeed, inhibition of cell growth starts with ATM auto-phosphorylation, which in turn leads to the phosphorylation and consequent activation of LKB1. Then, activated LKB1 phosphorylates AMPK on Thr172, which in turn phosphorylates TSC2 on Thr1227 and Ser1345 [10]. Interestingly, ATM can also phospho-activate AMPK at Thr172 independently of LKB1 [11,12,13]. AMPK regulates DDR, dictating the cell fate choice between autophagy and apoptosis [14]. Its inhibition induces DNA damage and apoptosis in cancer cells [15], while its knockout (KO) was found to cause increased susceptibility to apoptosis triggered by cisplatin, suggesting that AMPK negatively regulates cisplatin-induced apoptosis [16]. In addition, it was recently shown that AMPK phospho-activation induced by metformin-modified chitosan decreases PD-L1 expression in tumors, thereby inhibiting DNA repair upon cisplatin exposure [17]. More importantly, AMPK can suppress the mTOR signaling pathway by phospho-inactivating the key mTORC1 component RAPTOR on Ser722/Ser792. This phosphorylation induces cell cycle arrest when cells are damaged [18,19].

We recently showed that both mTOR and AMPK are novel interactors of the SMYD3 methyltransferase and validated these interactions in several gastrointestinal cancer cell lines [20]. Importantly, SMYD3 interacts with the active phosphorylated form of mTOR and AMPK [20]. SMYD3 methylates both histone and non-histone proteins, thereby orchestrating their interactions and functions, and is involved in cancer-related pathways [21]. Of note, it was recently found that SMYD3 is required for DNA repair [22]. In particular, its phosphorylation by ATM enables the formation of an ATM/SMYD3/CHK2/BRCA2 complex that is involved in DSB resolution [22]. Thus, here we investigated the role of SMYD3 in regulating the crosstalk between the mTOR and AMPK pathways upon exposure to DNA-damaging drugs.

## 2. Materials and Methods

### 2.1. Chemicals

Neocarzinostatin (#N9162), rapamycin (#R8781), and AICAR (#A9978) were purchased from Merck (Merck KGaA, Darmstadt, Germany). EM127 was synthesized as described in [17]. For each chemical, doses and treatment duration are indicated in the figure legends.

### 2.2. Cell Line Cultures

HCT-116, AGS, and MDA-MB-231 cell lines were purchased from ATCC. HCT-116 and MDA-MB-231 cells were cultured in DMEM high glucose (HG) without pyruvate (#11360-070, Gibco, Billings, MT, USA) with 10% FBS (#10270-106, Gibco, Billings, MT, USA) and 100 IU/mL penicillin-streptomycin (#15140-122, Gibco, Billings, MT, USA). AGS cells were cultured in RPMI HG without pyruvate (#21875-034, Gibco, Billings, MT, USA) with 10% FBS (#10270-106, Gibco, Billings, MT, USA) and 100 IU/mL penicillin-streptomycin (#15140-122, Gibco, Billings, MT, USA).

All cell lines were tested to be mycoplasma-free (#117048, Minerva Biolabs, Berlin, Germany) multiple times throughout the study. All cell cultures were maintained in a humidified incubator at 37 °C and 5% CO_2_.

### 2.3. Generation of SMYD3-KO Cell Lines

SMYD3-KO cell lines were generated using the CRISPR/Cas9 system. MDA-MB-231 SMYD3 KO cells were generated as previously described [16]. Briefly, they were transfected with the all-in-one expression vector Cas9-CD4+-SMYD3 gRNA using Lipofectamine 3000 (#L3000015, Thermo Fisher Scientific, Waltham, MA, USA) according to the manufacturer’s instructions. After 48 h, MDA-MB-231 CD4+ cells were enriched using the Dynabeads CD4 Positive Isolation Kit (#11331D, Thermo Fisher Scientific, Waltham, MA, USA) according to the manufacturer’s instructions. Isolation of clonal populations was performed with agarose-based cloning rings (#C1059, Merck KGaA, Darmstadt, Germany). Cell clones were tested for site-specific loss of function alterations by PCR, using the following sequencing primers: SMYD3 gRNA FW 5′AGCCCGTGAGACGCCCGCTGCTGG and SMYD3 gRNA RV 5′GAAAAGTTCGCAACCGCCAA. Sequencing products were purified using the Dye Ex 2.0 Spin Kit (#63204, QIAGEN, Hilden, Germany) and sequenced on an ABI PRISM 310 Genetic Analyzer (Applied Biosystems, Waltham, MA, USA).

HCT-116 cells were transfected with the TrueCut Cas9 Protein V2 and SMYD3 TrueGuide gRNAs (#CRISPR1032607 and #CRISPR1032618, Invitrogen, Carlsbad, CA, USA) using Lipofectamine CRISPRMAX transfection reagent (#CMAX00001, Invitrogen, Carlsbad, CA, USA) according to the manufacturer’s instructions. After 48 h, the isolation of clonal populations was performed with agarose-based cloning rings (#C1059, Merck KGaA, Darmstadt, Germany). Cell clones were tested for site-specific loss of function alterations by PCR, using the following sequencing primers: SMYD3 gRNA FW 5’AGCCCGTGAGACGCCCGCTGCTGG and SMYD3 gRNA RV 5’GAAAAGTTCGCAACCGCCAA. Sequencing products were purified using the Dye Ex 2.0 Spin Kit (#63204, QIAGEN, Hilden, Germany) and sequenced on an ABI PRISM 310 Genetic Analyzer (Applied Biosystems, Waltham, MA, USA).

### 2.4. Co-Immunoprecipitation Assays

Cells treated or not with NCS, as indicated in the figure legends, were collected and homogenized in lysis buffer (50 mM Tris-HCl pH 7,4, 5 mM EDTA, 250 mM NaCl, and 1% Triton X-100) supplemented with protease and phosphatase inhibitors (Roche, Basel, Switzerland). Coupling between Dynabeads Protein A (#10002D, Thermo Fisher Scientific, Waltham, MA, USA) and antibodies, i.e., anti-SMYD3 (#12859, Cell Signaling Technology, Danvers, MA, USA), anti-AMPK (#2532, Cell Signaling Technology, Danvers, MA, USA), or anti-mTOR (#2972, Cell Signaling Technology, Danvers, MA, USA), was performed in 100 μL of 0.01% Tween 20-1X PBS for 45 min at room temperature on a rocking platform. Cell lysates were immunoprecipitated with antibody-bead complexes. Immunocomplexes were washed extensively, boiled in Laemmli sample buffer, and subjected to SDS-PAGE and immunoblot analysis. IgGs were used as a negative control. Primary antibodies used: anti-SMYD3 (#12859, Cell Signaling Technology, Danvers, MA, USA), anti-AMPK (#2532, Cell Signaling Technology, Danvers, MA, USA), and anti-mTOR (#2972, Cell Signaling Technology, Danvers, MA, USA). Mouse anti-rabbit IgG (Conformation Specific) HRP (#5127, Cell Signaling Technology, Danvers, MA, USA) was used as a secondary antibody and revealed using the ECL plus chemiluminescence reagent (#RPN2232, GE Healthcare, Chicago, IL, USA) according to the manufacturer’s instructions.

### 2.5. Immunoblotting

Whole-cell extracts were obtained from cells collected and homogenized in lysis buffer (50 mM Tris-HCl pH 7.4, 5 mM EDTA, 250 mM NaCl, and 1% Triton X-100) supplemented with protease and phosphatase inhibitors (Roche, Basel, Switzerland). A total of 20 μg of protein extract from each sample was denatured in Laemmli sample buffer and loaded into an SDS-poly-acrylamide gel for immunoblot analysis. Primary antibodies used: anti-Acetyl-CoA Carboxylase (#3676, Cell Signaling Technology, Danvers, MA, USA), anti-phospho-Acetyl-CoA Carboxylase (Ser79) (#3661, Cell Signaling Technology, Danvers, MA, USA), anti-AMPK-α (#2532, Cell Signaling Technology, Danvers, MA, USA), anti-phospho-AMPK-α (Thr172) (#2531, Cell Signaling Technology, Danvers, MA, USA), anti-phospho-Histone H2A.X (Ser139) (#9718, Cell Signaling Technology, Danvers, MA, USA), anti-p70 S6 Kinase (#34475, Cell Signaling Technology, Danvers, MA, USA), anti-phospho-p70 S6 Kinase (Thr389) (#9234, Cell Signaling Technology, Danvers, MA, USA), anti-RAPTOR (#2280, Cell Signaling Technology, Danvers, MA, USA), anti-phospho-RAPTOR (Thr792) (#89146, Cell Signaling Technology, Danvers, MA, USA), anti-S6 Ribosomal Protein (#2217, Cell Signaling Technology, Danvers, MA, USA), anti-phospho-S6 Ribosomal Protein (Ser240/244) (#5364, Cell Signaling Technology, Danvers, MA, USA), anti-VINCULIN (#13901, Cell Signaling Technology, Danvers, MA, USA). Anti-rabbit IgG HRP and anti-Mouse IgG HRP (#NA934V and #NA931V, respectively, Merck KGaA, Darmstadt, Germany) were used as secondary antibodies and revealed using the ECL-plus chemiluminescence reagent (#RPN2232, Merck KGaA, Darmstadt, Germany) according to the manufacturer’s instructions. Densitometric evaluation was performed using ImageJ software (version 1.45, National Institutes of Health, Bethesda, Maryland, USA)

### 2.6. In Vitro Methylation Assay

Analysis of SMYD3 methylation activity was performed using a luminometric methylation assay (MTase-Glo™ Methyltransferase Assay, #V7601, Promega, Madison, WI, USA). Briefly, SMYD3 active protein (500 ng, S348-380CG SignalChem, Richmond, BC, Canada) was assayed in a methylation reaction buffer containing 50 mM Tris (pH 8), 4 mM MgCl_2_, 0.2% Tween-20, 2 mM dithiothreitol (DTT), 200 μM SAM, and 500 ng of AMPK protein (P48-14H-20, Signalchem, Richmond, BC, Canada) in a final volume of 20 μL. The reaction was incubated overnight at 30 °C. Then, 5 μL of 5× MTase-Glo reagent were added to convert SAH to ADP. Next, MTase-Glo™ Detection Solution was added to convert ADP to ATP, which was determined by a luciferase/luciferin reaction. The generated luminescence was measured using a luminometer (SPECTROstar Omega, BMG LABTECH, Ortenberg, Germany). Each data point was collected in triplicate.

### 2.7. In Silico Methylation Prediction Analysis

To identify the putative SMYD3 lysine methylation sites in AMPK sequence, we performed an in silico methylation prediction analysis with three different servers: GPS-MSP (http://msp.biocuckoo.org/; cutoff 0.5, accessed on 20 May 2023), Musite Deep (https://www.musite.net/; cutoff 0.2 accessed on 20 May 2023), and MethylSight https://methylsight.cu-bic.ca/, cutoff 0.5 accessed on 20 May 2023).

### 2.8. Mass Spectrometry Analysis

Mass spectrometry (MS) analysis was performed by the Cogentech SRL service. Gel bands were subjected to reduction with DTT (#A39255, Thermo Fisher Scientific, Waltham, MA, USA), alkylation with iodoacetamide (IAA) (#A39271, Thermo Fisher Scientific, Waltham, MA, USA), and digestion with trypsin (#90059, Thermo Fisher Scientific, Waltham, MA, USA). Then, the flow-through was treated with C18 Spin Tips & Columns (#84850, Thermo Fisher Scientific, Waltham, MA, USA) for desalting. The samples and the desalted flow-through were further purified with SP3 and then analyzed by nLC-ESI-MS/MS on a Q Exactive HF mass spectrometer (Thermo Fisher Scientific, Waltham, MA, USA) with a 45-min gradient. Samples were run in technical duplicate in a positive mode with electrospray ionization. Data acquisition and processing were performed with Analyst TF (version 1.7.1, AB SCIEX, Singapore). Data were analyzed using the Proteome Discoverer (version 2.5, Thermo Fisher Scientific, Waltham, MA, USA), Mascot(version 2.8, Matrix Science Inc., Boston, MA, USA),and Scaffold (version 5.2.2, Proteome Software Inc., Portland, OR, USA) setting software. The parameter settings for data processing were as follows: DataBase = Uniprot_CP_Human_2020_AMPK A2/B1/G1 (Database Uniprot_cp_Human + AMPK A2/B1/G1; Human sequence, Accession Numbers: P54646/Q9Y478/P54619); Enzyme = Trypsin (cuts at C-term of K and also on R); Modifications = Acetyl (Protein N-term), Carbamidomethyl (C), Oxidation (M), Phosphorylation (STY); Peptide Thresholds: 95.0% minimum; Protein Thresholds: 99.0% minimum; 2 peptides/protein minimum.

### 2.9. Multiple Sequence Alignment

The protein sequences of human AAPK2 and homologous proteins from other species were aligned with the latest version of Clustal Omega (http://www.clustal.org/, accessed on 20 May 2023), an online tool allowing to align sequences with high accuracy.

### 2.10. Statistical Analysis

Statistical analysis was performed using Student’s *t*-test. Differences were considered significant when the *p*-value was <0.05. At least three independent experiments were performed for each assay.

## 3. Results

### 3.1. SMYD3 Is Involved in AMPK-mTOR Signaling Balance during DDR

Emerging evidence suggests that mTOR and AMPK signaling pathways are involved in DDR [6,14]. Therefore, we investigated the modulation of these cascades in response to the DNA-damaging agent NCS by evaluating several of their downstream effectors by immunoblot analysis. To this end, we used HCT-116 colorectal cancer, AGS gastric cancer, and MDA-MB-231 breast cancer cells, in which we previously showed that SMYD3 is required for DNA repair upon NCS exposure [22]. After 24 h treatment with NCS, we detected increased levels of phospho-proteins involved in the AMPK pathway and decreased levels of phospho-proteins involved in the mTOR pathway in all cell lines (Figure 1a). Specifically, besides phospho-activation of AMPK itself, we observed upregulation of the phosphorylated form of acetyl-CoA carboxylase (ACC), an AMPK-specific downstream target that was previously shown to be activated in a dose-dependent manner upon IR exposure [23]. In addition, we evaluated RAPTOR phosphorylation by AMPK at Ser722/Ser792 [18] and found that it increased upon NCS exposure (Figure 1a). This phosphorylation event has been reported to be essential for the inhibition of mTORC1 and for the downregulation of its signaling cascade, resulting in cell cycle arrest under energy-stress conditions [18]. Of note, RAPTOR phospho-modulation can be considered a critical switch between the AMPK and mTOR signaling pathways. For what concerns the mTOR cascade, we investigated the phosphorylation of its downstream target p70S6K at threonine 389, which was previously reported to decrease upon exposure to DNA-damaging drugs [24]. Our results showed that NCS reduced the phosphorylated form of p70S6K (Figure 1a). Consistently, we observed that phosphorylation of the S6 ribosomal protein, a direct mTOR downstream target previously found to be involved in the regulation of DNA repair [25], at serines 240/244 also decreased in NCS-treated cancer cells (Figure 1a).

Subsequently, we explored the role of SMYD3 in the regulation of the AMPK and mTOR pathways. To this end, we used the SMYD3-KO MDA-MB-231 cell line, which we had previously engineered and analyzed to assess SMYD3 involvement in DNA damage [22], and we generated a SMYD3-KO HCT-116 cell line with the CRISPR-Cas9 system for genome editing. Then, we evaluated the response of these cells to NCS treatment. Interestingly, in contrast to what we observed in the respective parental cell lines, we did not find a significant modulation of the AMPK and mTOR pathways in SMYD3-KO cancer cells during DDR, as assessed by changes in the expression levels of relevant phospho-activated downstream targets (Figure 1b).

These data suggest that SMYD3 plays a key role in regulating the balance between the AMPK and mTOR pathways during DDR.

### 3.2. Pharmacological Inhibition of SMYD3 Affects the AMPK and mTOR Pathways

To further elucidate SMYD3 function in the crosstalk between its binding partners AMPK and mTOR upon NCS exposure, we treated HCT-116 and AGS gastrointestinal cancer cells and the MDA-MB-231 breast cancer cell line with the novel active site-selective covalent SMYD3 inhibitor EM127. This 4-aminopiperidine derivative bears a 2-chloroethanoyl group acting as a selective reactive site for the Cys186 residue located in the substrate/histone binding pocket of SMYD3 [26]. Intriguingly, our results showed that treating cells with 5 μM EM127 for 24 h to pharmacologically inhibit SMYD3 reduced the amount of the phosphorylated form of AMPK and its downstream targets ACC and RAPTOR (Figure 2a–c), while increasing p70S6K and S6 phospho-activation (Figure 2a–c). Moreover, cells pre-treated with 5 μM EM127 for 24 h and subsequently exposed to NCS for 24 h failed to correctly activate the AMPK pathway, while showing activation of the mTOR cascade (Figure 2a–c).

These data indicate that SMYD3 enzymatic activity is required for the modulation of these two pathways during DDR.

### 3.3. SMYD3 Bridges the Gap between the AMPK and mTOR Cascades upon NCS Exposure

To further assess the role of SMYD3 in the modulation of AMPK signaling, we evaluated the effect of EM127 pre-treatment on AMPK activation by AICAR, an adenosine analog known to specifically trigger the AMPK pathway [27]. Our results showed that treating HCT-116 cells with EM127 prevented AMPK activation by AICAR, as revealed by decreased phosphorylation of AMPK, ACC, and RAPTOR (Figure 3a). Of note, activation of the AMPK cascade was also affected by SMYD3 inhibition upon AICAR+NCS combined treatment (Figure 3a). These findings indicate that AICAR does not bypass the effect of SMYD3 inhibition on the activation of the AMPK pathway and confirm that SMYD3 enzymatic activity plays a major role in this process. Then, to establish whether the effect of SMYD3 on p70S6K activation and therefore S6 phosphorylation is mediated by mTOR, we treated HCT-116 cells with the mTOR inhibitor rapamycin. Our results showed that rapamycin drastically reduced the levels of the phosphorylated form of both p70S6K and S6. Importantly, EM127 pre-treatment did not promote p70S6K and S6 phospho-activation upon rapamycin inhibition (Figure 3b). These results were not affected by concomitant exposure to NCS (Figure 3b). These findings revealed that regulation of p70S6K activation by SMYD3 is mTOR-dependent.

### 3.4. SMYD3 Forms a Multiprotein Complex with AMPK and mTOR

Previous studies on AMPK/mTOR crosstalk revealed that these proteins interact directly with each other [28]. As we recently showed that SMYD3 can interact with mTOR and AMPK, we hypothesized that SMYD3 may participate in a multiprotein complex containing both AMPK and mTOR. To ascertain whether this occurs in our cellular system, we performed co-immunoprecipitation experiments by pulling down SMYD3, AMPK, or mTOR in HCT-116 and MDA-MB-231 cells exposed or not to NCS. Our results showed that SMYD3 is part of a multiprotein complex that also comprises AMPK and mTOR and is involved in DDR (Figure 4).

### 3.5. SMYD3 Can Directly Methylate AMPK

The above data indicate that SMYD3 enzymatic activity is crucial for the regulation of the AMPK cascade. These findings prompted us to better characterize the mechanism of action of SMYD3 in this process by performing an in vitro methylation assay. Our results showed that SMYD3 can directly methylate AMPK (Figure 5a). In order to identify the AMPK residues that may potentially be methylated by SMYD3, we performed an in silico methylation prediction analysis with three different servers (GPS-MSP, Methyl Sight, and Musite Deep). These servers consider various sequence and structural features of the proteins (i.e., the secondary structure of residues surrounding methylation sites, methylated lysines previously assessed in in vivo studies, and accessible surface area). Methyl Sight identified the highest number of putative lysine methylation sites, including those predicted by the two other tools. This server uses a new machine learning model developed to find lysine methylation sites based on previously assessed MS findings in in vivo proteomic studies. Based on this in silico analysis, the AMPK residues that may potentially be methylated by SMYD3 include Lys4, Lys6, Lys12, Lys29, Lys31, Lys 41, Lys45, Lys51, Lys53, Lys60, Lys62, Lys141, Lys154, Lys364, Lys379, Lys391, Lys393, Lys398, Lys399, Lys401, Lys411, Lys424, Lys431, Lys452, and Lys470 (Figure 5b). To confirm which AMPK lysine residues are targeted by SMYD3, we performed an MS analysis of an AMPK recombinant protein (AMPK A2/B1/G1) after methylation by SMYD3 in vitro. The two peptides SQSK411PYDIMAEVYR and AMK424QLDFEWK, obtained by double proteolytic digestion with the endoproteinases trypsin and Glu-C, showed a mass increase of 42 Da each, which corresponds to the weight of an additional methyl group at Lys411 and Lys424 (Figure 5c). In order to evaluate the evolutionary and functional relevance of these lysine residues, we carried out a multiple alignment of AAPK2 (AMPK catalytic subunit alpha-2) and homologous proteins from various species, ranging from Caenorhabditis elegans to humans. Interestingly, lysines 411 and 424 are found in regions that are highly conserved in human AAPK2 and homologous proteins from other species (Figure 5d). This is consistent with the evidence that post-translational modifications preferentially occur in evolutionarily conserved regions.

Overall, our findings indicate that AMPK methylation by SMYD3 may positively regulate its signaling pathway and therefore AMPK/mTOR crosstalk during DDR.

## 4. Discussion

The AMPK and mTOR cascades are critical signaling pathways that coordinate many cellular processes in response to stress events inducing metabolic changes and DNA damage [6,14,29]. These pathways share various key components that cooperate to overcome the damage status [18,30]. In particular, upon DNA damage, AMPK-mTOR crosstalk leads to a pro-survival response as a result of AMPK activation, which negatively regulates mTOR [31]. While this can be beneficial to cells in several circumstances, in cancer, it may promote oncogenic progression. Indeed, through this crosstalk, cancer cells can delay or arrest cell cycle progression to allow the resolution of damaged DNA by activating the DNA repair machinery [32].

AMPK has been shown to act as a regulator of DNA repair systems and other processes determining cell fate [14]. In particular, AMPK seems to play a major role in balancing energy homeostasis and cell survival in DDR [33,34,35,36]. Although the molecular mechanisms underlying AMPK involvement in DNA damage have not been fully elucidated yet, AMPK has been reported to phosphorylate TSC2 at Thr1227 and Ser1345, which leads to RHEB inhibition and mTOR pathway blockage [13,37]. The PIKK mTOR is evolutionarily connected with other PIKKs that are involved in DDR, such as ATM [6], and their interconnections lead to a number of negative and positive feedback loops that regulate both cell cycle progression and DNA repair [8].

Our results identified SMYD3 as a key player in the regulation of AMPK-mTOR signaling balance during DDR. Interestingly, SMYD3 pharmacological inhibition prevented the activation of the AMPK cascade and the downregulation of mTOR signaling upon NCS-induced DNA DSBs. This is expected to reverse the pro-survival response usually mediated by these pathways, shifting cell fate toward cell death.

Recent evidence suggests that SMYD3 can orchestrate important cellular processes, including DDR, by enabling the formation/activation of functional multiprotein complexes that mediate the propagation of signaling cascades [22]. In line with this view, here we showed that SMYD3 is part of a multiprotein complex comprising AMPK and mTOR in cancer cells exposed to DNA-damaging drugs. Of note, we also found that SMYD3 methylates AMPK in vitro at Lys411 and Lys424, which belong to an evolutionarily conserved region and are thus likely to be involved in an essential function or mediate an important structural characteristic [38]. As such, it can be speculated that this SMYD3-mediated AMPK post-translational modification may be required for the modulation of the activity of the multiprotein complex comprising SMYD3, AMPK, and mTOR and therefore play a role in the regulation of AMPK-mTOR signaling balance. This hypothesis will need to be confirmed in future studies.

Since AMPK is a key player in the modulation of various cancer-related processes, including DDR, it is considered an appealing target for cancer therapy. In this light, our results suggest that targeting the SMYD3 methyltransferase to disrupt the signaling balance between AMPK and mTOR in cancer cells may represent a suitable therapeutic approach.

The SMYD3 interactor mTOR is involved in the “avoiding immune destruction” cancer hallmark [20], and its signaling is a key regulator of immune cell metabolism and function [39]. AMPK also plays an important role in the regulation of the anti-tumor immune response; in particular, it can act as a regulator of PD-L1 by reducing its expression [17], which is a promising strategy to reactivate immunotherapy [40]. Moreover, recent studies identified SMYD3 as a mediator of immune escape in human papilloma virus (HPV)-negative head and neck squamous cell carcinoma (HNSCC). Its depletion promotes the influx of CD8+ T cells and increases sensitivity to PD-1 inhibitors, thereby suggesting future translational approaches combining SMYD3 inhibition with checkpoint blockade to obtain better outcomes [41]. Based on our findings on the role played by SMYD3 in the modulation of AMPK-mTOR signaling balance, it can be speculated that SMYD3 activity on AMPK might promote the downregulation of PD-L1, thereby suggesting a role in the modulation of AMPK and mTOR signaling balance in the context of cancer immunosurveillance. Considering that immunotherapy is among the most promising new cancer treatments, confirming the potential of SMYD3 pharmacological blockade may pave the way toward effective translational advances.

## 5. Conclusions

Further investigations in preclinical models are needed to validate the potential of this strategy in gastrointestinal and breast cancers and to translate these findings into therapeutic opportunities. To this end, challenges and possible obstacles related to the design of SMYD3 inhibitors should be evaluated. Selectivity and specificity are two crucial properties that have to be tested to avoid off-target effects and minimize potential toxicity, as well as its bioavailability and metabolic stability. In this case, drug delivery systems and nanotechnology may be considered for targeted delivery and improved therapeutic efficacy of SMYD3 inhibitors. Hence, future studies should investigate the effect of SMYD3 inhibitors, with the aim of developing an effective and safe strategy that can be assessed in clinical trials, thereby guiding the future directions of this treatment strategy.

## Figures and Tables

**Figure 1 cells-12-02644-f001:**
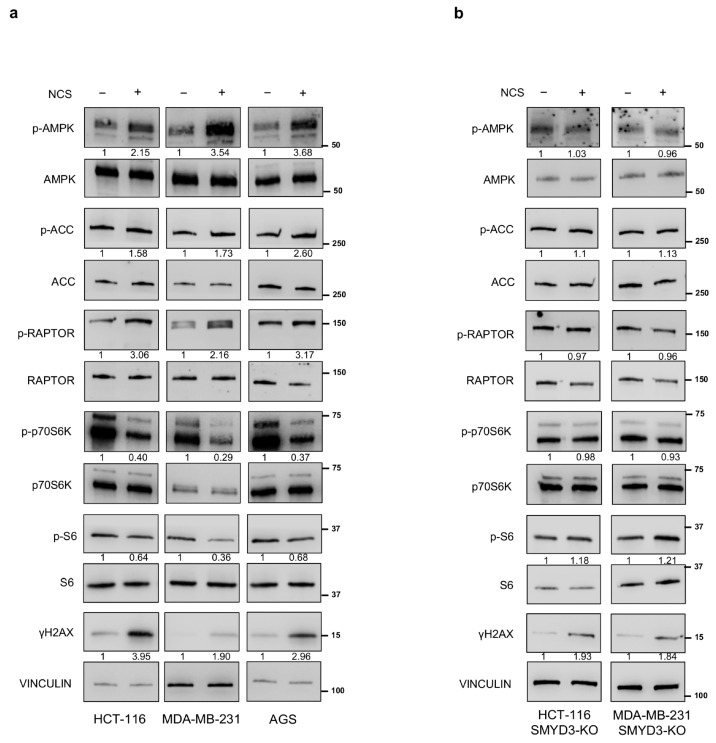
Effects of NCS on the AMPK/mTOR pathways. (**a**) Immunoblotting showing the levels of proteins involved in the AMPK and mTOR pathways in HCT-116, MDA-MB-231, and AGS cells treated for 24 h with NCS (5 nM). (**b**) Immunoblotting showing the levels of proteins involved in the AMPK and mTOR pathways in SMYD3-KO cells (HCT-116 and MDA-MB-231) treated for 24 h with NCS (5 nM). H2AX phosphorylation (γH2AX) was analyzed as a control of the induced-DNA damage, VINCULIN was used as a loading control.

**Figure 2 cells-12-02644-f002:**
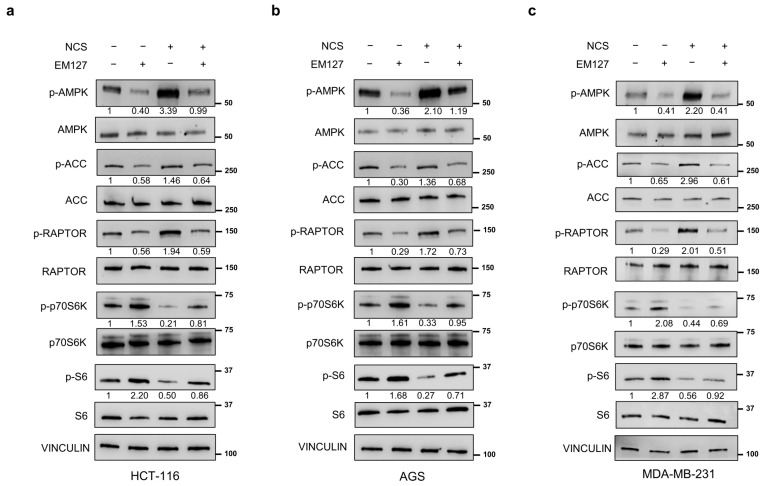
Effects of NCS on the AMPK/mTOR pathways after pre-treatment with EM127. (**a**–**c**) Immunoblotting showing the levels of proteins involved in the AMPK and mTOR pathways in HCT-116 (**a**), AGS (**b**) and MDA-MB-231 (**c**) cells pre-treated with the SMYD3 inhibitor EM127 (5 μM) for 24 h and then treated with NCS (5 nM) for 24 h. VINCULIN was used as a loading control.

**Figure 3 cells-12-02644-f003:**
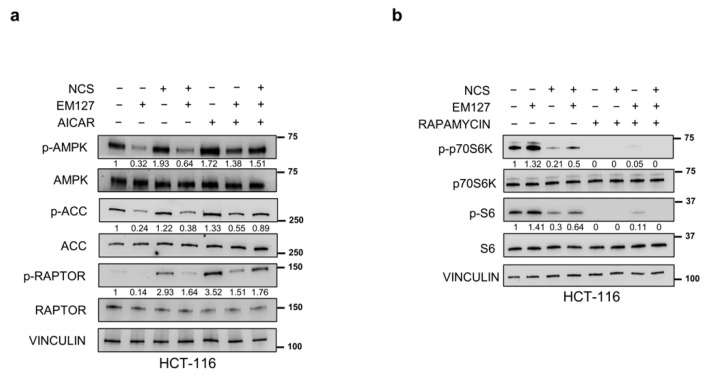
Role of SMYD3 in the modulation of AMPK and mTOR signaling pathways. (**a**) Immunoblotting showing the levels of proteins involved in the AMPK pathway in HCT-116 cells pre-treated or not with EM127 (5 μM) for 24 h and then treated with NCS (5 nM) for 24 h and/or AICAR (5 mM) for 24 h. (**b**) Immunoblotting showing the levels of proteins involved in the mTOR pathway in HCT-116 cells pre-treated or not with EM127 (5μM) for 24 h and then treated with NCS (5 nM) for 24 h and/or rapamycin (100 nM) for 4 h. VINCULIN was used as a loading control.

**Figure 4 cells-12-02644-f004:**
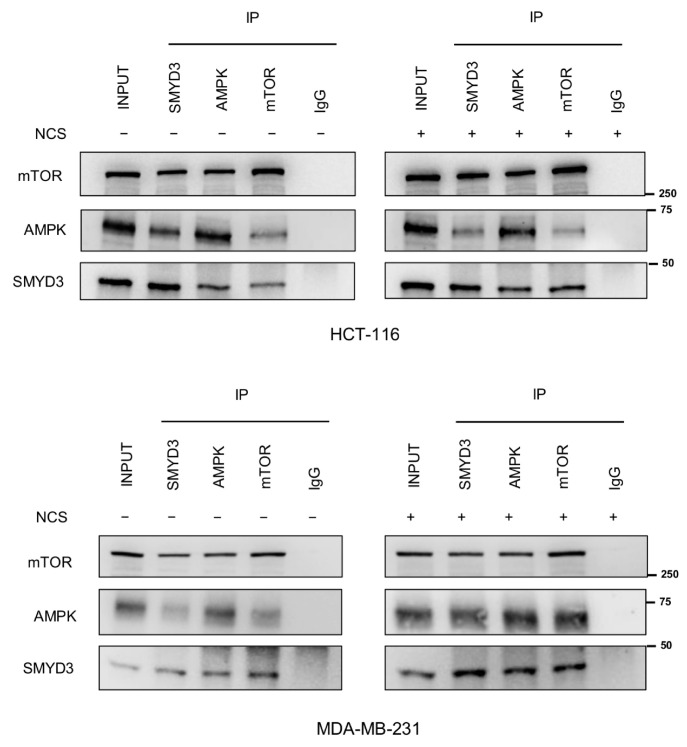
Functional interaction between SMYD3, AMPK, and mTOR. Co-immunoprecipitation of endogenous SMYD3, AMPK, or mTOR in HCT-116 and MDA-MB-231 cells treated or not with NCS (5 nM) for 24 h. Anti-IgGs were used as a negative control.

**Figure 5 cells-12-02644-f005:**
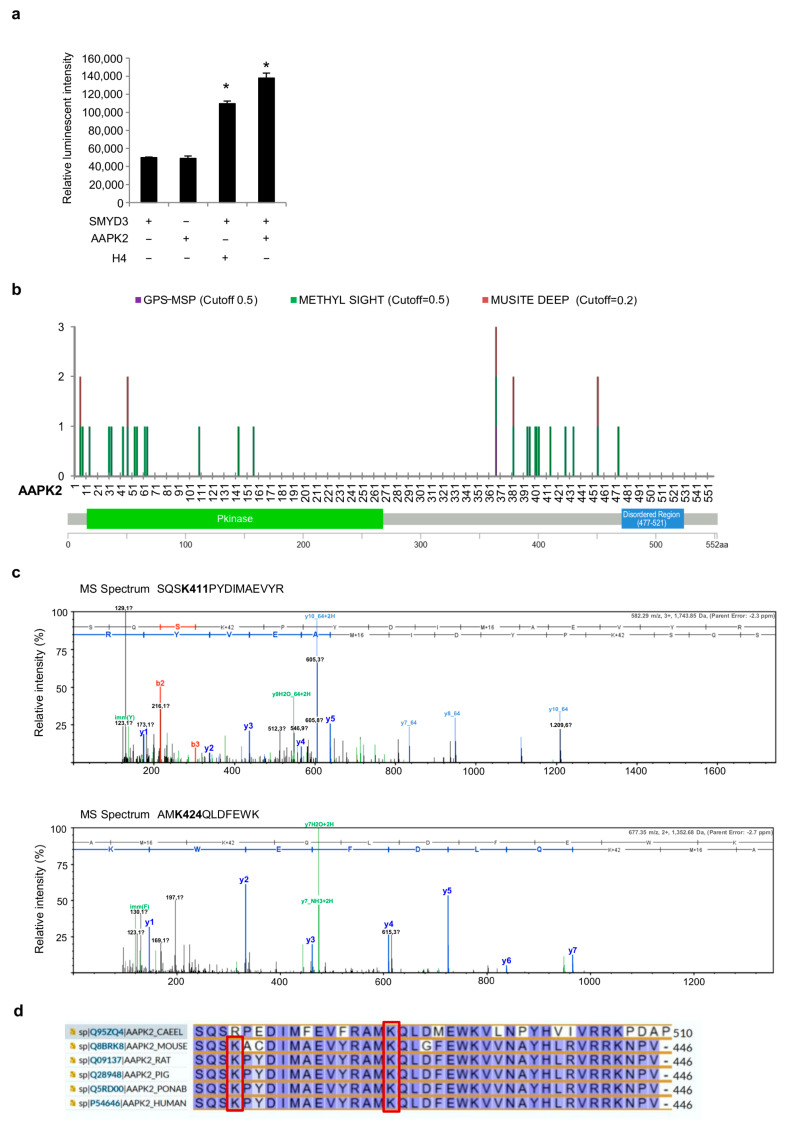
SMYD3 methylates AMPK. (**a**) In vitro methylation assay showing AMPK (AAPK2) methylation by SMYD3. H4 was used as a SMYD3 control substrate. * *p* < 0.05 vs active SMYD3 (**b**) In silico methylation prediction analysis. Three in silico prediction servers were used to identify AMPK consensus methylation sites: GPS-MSP, Methyl Sight, and Musite Deep. (**c**) MS/MS spectrum of SQSK411PYDIMAEVYR and AMK424QLDFEWK, two peptides obtained by double proteolytic digestion of SMYD3-methylated AMPK with the endoproteinases trypsin and Glu-C. (**d**) Multiple sequence alignment of human AAPK2 and homologous proteins from other species. UniProt IDs are indicated on the left. Lysines 411 and 424 (red boxes) are located in highly conserved regions. CAEEL: *C. elegans*, PONAB: *Pongo abelii*.

## Data Availability

Data are contained within the article.

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
