# Peer review of "SMYD3 Modulates AMPK-mTOR Signaling Balance in Cancer Cell Response to DNA Damage"

_cells, 2023, doi:10.3390/cells12222644_

Round 1

Reviewer 1 Report

Comments and Suggestions for Authors

M. Lepore Signorile et al claim that  in gastric and breast cell lines SMYD3 enables the formation of a multiprotein complex comprising AMPK and mTOR during DNA damage response to the double-strand break-inducing agent neocarzinostatin. SMYD3 methylates in vitro AMPK and SMYD3 inhibition actives mTOR through the inhibition of AMPK.

Major points

1.      In the abstract authors claim that the effect they observe is in  both gastric and breast cancer cell lines, but actually most of the data are presented only  for gastric cell lines, but for breast cell lines only the first experiment is shown. The authors should include the modulation by SYMD3i and the multicomplex formation also in breast cell line in order to include in the abstract.

2.      In figure  4 the authors show by IP de formation of a multiprotein complex between SYMD3, AMPK and mTOR. This complex is formed independently of the treatment with NCS, Authors should argue on that as they claim the SYMD3 is recruited to the complex after DNA damage

3.      In some parts of the text the authors generalized they observation to any DNA damage, but actually they only show the effect with neocrazinostatin, this should be state in the text.

Minor  points

In all the figures for some biomarkers is difficult to see an increase or decrease of its phosphorylated form. A quantification of the bands should be included.

Author Response

Dear Editor,

we are pleased to submit the amended version of our work “SMYD3 modulates AMPK-mTOR signaling balance in cancer cell response to DNA damage” (cells-2681387), which we would like to have considered for publication in Cells as part of the special issue “DNA Damage and DNA Repair: What’s New in Biology and Cancer Therapeutics?”. We addressed below all the comments raised by the Reviewers, mainly by performing new experiments and by responding/clarifying or adding novel sentences in the text.

REVIEWER 1

Lepore Signorile et al claim that  in gastric and breast cell lines SMYD3 enables the formation of a multiprotein complex comprising AMPK and mTOR during DNA damage response to the double-strand break-inducing agent neocarzinostatin. SMYD3 methylates in vitro AMPK and SMYD3 inhibition actives mTOR through the inhibition of AMPK

Major points

  1. In the abstract authors claim that the effect they observe is in  both gastric and breast cancer cell lines, but actually most of the data are presented only  for gastric cell lines, but for breast cell lines only the first experiment is shown. The authors should include the modulation by SYMD3i and the multicomplex formation also in breast cell line in order to include in the abstract.

We thank the Reviewer for this comment. We have previously showed the effect of SMYD3 inhibition and the formation of the multiprotein complex in gastrointestinal cancer cell lines. Thus, we have performed the analysis in the breast cancer cell line MDA-MB-231 which are now depicted in Figure 2c and Figure 4.

  1. In figure  4 the authors show by IP de formation of a multiprotein complex between SYMD3, AMPK and mTOR. This complex is formed independently of the treatment with NCS, Authors should argue on that as they claim the SYMD3 is recruited to the complex after DNA damage

We thank the Reviewer for this comment. We mistakenly reported that SMYD3 enables the formation of the multiprotein complex during DNA damage response. Our intent was to show that SMYD3 is part of a multiprotein complex that also comprises AMPK and mTOR and to convey the idea that its involvement in the complex promotes its activity in response to DNA damage. Our subsequent speculation, reported in the discussion, was that SMYD3-mediated AMPK methylation could be required for the modulation of the activity of this multiprotein complex and therefore play a role in the regulation of AMPK-mTOR signaling balance during DNA damage. In this amended version of the manuscript, we corrected this information.

  1. In some parts of the text the authors generalized they observation to any DNA damage, but actually they only show the effect with neocrazinostatin, this should be state in the text.

We thank the Reviewer for this comment. As reported in the manuscript, neocarzinostatin is a radiomimetic drug that causes double-strand breaks in the DNA. In this amended version of the manuscript, this is explicitly specified.

Minor  points

In all the figures for some biomarkers is difficult to see an increase or decrease of its phosphorylated form. A quantification of the bands should be included.

We thank the Reviewer for this suggestion. In this amended version of the manuscript, we added the relative quantification of immunoblotting bands to better highlight the increase or decrease of the phosphorylated form of the analyzed proteins in the different experimental conditions used in the study.

Reviewer 2 Report

Comments and Suggestions for Authors

      In this research, the authors revealed that SMYD3 modulated AMPK-mTOR signaling balance in cancer cell response to DNA damage. In my opinion, the current version of this manuscript fits the scope of Cells and could be accepted after minor revision.

My specific comments are in detail listed below:

1.     Some references are out of date (before 2010). Some new recent references may be better.

2.     As recently proved, AMPK played a vital role in in cancer cell response to DNA damage by regulating PD-L1 expression. In my opinion, the authors should discuss it clear in the introduction (Line 34-65). Some references should be added to this part including 10.1016/j.ijbiomac.2022.10.167.

3.     Some English usage needs polish in this paper. The authors should check it carefully.

4.     In the discussion part, the authors may predict or discuss how SMYD3 affect the immune status of tumors, especially the expression of PD-L1 since enhanced AMPK phosphorylation may decrease PD-L1 expression in tumors. Some references should be added to this part including 10.1002/adma.202206121.

5.     The results of western blot assay should be quantified by Image J or some other related softwares to show a more clear results.

6.     In the conclusion part, the authors may predict the directions and obstacles of using SMYD3 to affect cancer cell response.

Author Response

Dear Editor,

we are pleased to submit the amended version of our work “SMYD3 modulates AMPK-mTOR signaling balance in cancer cell response to DNA damage” (cells-2681387), which we would like to have considered for publication in Cells as part of the special issue “DNA Damage and DNA Repair: What’s New in Biology and Cancer Therapeutics?”. We addressed below all the comments raised by the Reviewers, mainly by performing new experiments and by responding/clarifying or adding novel sentences in the text.

REVIEWER 2

In this research, the authors revealed that SMYD3 modulated AMPK-mTOR signaling balance in cancer cell response to DNA damage. In my opinion, the current version of this manuscript fits the scope of Cells and could be accepted after minor revision.

We thank the Reviewer for this general comment. In this amended version of the manuscript, we addressed the observations as detailed in the below point-by-point response.

My specific comments are in detail listed below:

  1. Some references are out of date (before 2010). Some new recent references may be better.

We thank the Reviewer for this comment. In this amended version of the manuscript, whenever possible, we replaced older references with more recent publications or added more recent publications.

  1. As recently proved, AMPK played a vital role in in cancer cell response to DNA damage by regulating PD-L1 expression. In my opinion, the authors should discuss it clear in the introduction (Line 34-65). Some references should be added to this part including 10.1016/j.ijbiomac.2022.10.167.

We thank the Reviewer for this comment. As suggested, in this amended version of the manuscript, we extended the introduction by outlining the role of AMPK in cancer cell response to DNA damage, including its involvement in the regulation of PD-L1 expression, and added relevant references.

  1. Some English usage needs polish in this paper. The authors should check it carefully.

We thank the Reviewer for this suggestion. In this amended version of the manuscript, we revised the language and refined some sentences to improve clarity, and we carefully reviewed the manuscript to fix any language issues.

  1. In the discussion part, the authors may predict or discuss how SMYD3 affect the immune status of tumors, especially the expression of PD-L1 since enhanced AMPK phosphorylation may decrease PD-L1 expression in tumors. Some references should be added to this part including 10.1002/adma.202206121.

We are grateful to the reviewer for this suggestion. As suggested, in this amended version of the manuscript, we discussed potential implications of our findings in the context of the role played by SMYD3 in the immune status of tumors. Recent studies showed that SMYD3 can affect cancer immune response. In particular, it has been identified as a mediator of immune escape in human papillomavirus (HPV)-negative head and neck squamous cell carcinoma (HNSCC). Its depletion promotes the influx of CD8+ T cells and increases sensitivity to PD-1 inhibitors, thereby suggesting future translational approaches combining SMYD3 inhibition with checkpoint blockade to obtain better outcomes (Nigam et al., 2023). Moreover, the SMYD3 interactor mTOR is involved in the “avoiding immune destruction” cancer hallmark (Fasano et al., 2022), and its signaling is a key regulator of immune cell metabolism and function. AMPK also plays an important role in the regulation of anti-tumor immune response; in particular, it can act as a PD-L1 downregulator (Zhou et al., 2023). Thus, SMYD3 activity on AMPK might promote the downregulation of PD-L1 and therefore modulate both AMPK and mTOR signaling in the context of cancer immunosurveillance.

  1. The results of western blot assay should be quantified by Image J or some other related softwares to show a more clear results.

We thank the Reviewer for this suggestion. In this amended version of the manuscript, we added the relative quantification of immunoblotting bands to better highlight the increase or decrease of the phosphorylated form of the analyzed proteins in the different experimental conditions used in the study.

  1. In the conclusion part, the authors may predict the directions and obstacles of using SMYD3 to affect cancer cell response.

We thank the Reviewer for this suggestion. In this amended version of the manuscript, we added future research directions and obstacles of using SMYD3 inhibition as a therapeutic strategy for cancer treatment.

Round 2

Reviewer 1 Report

Comments and Suggestions for Authors

The authors have addressed all my suggestions, Then I recommend to accept the publication